# Preventing spread of aerosolized infectious particles during medical procedures: A lab-based analysis of an inexpensive plastic enclosure

Luke W. Monroe[1], Jack S. Johnson[1], Howard B. Gutstein[2], John P. Lawrence[2], Keith Lejeune[2], Ryan C. Sullivan[1]*, Coty N. Jen[1]

**1** Center for Atmospheric Particle Studies, Carnegie Mellon University, Pittsburgh, PA, United States of America, **2** Anesthesiology Institute, Allegheny Health Network, Pittsburgh, PA, United States of America

* rsullivan@cmu.edu

## Abstract

Severe viral respiratory diseases, such as SARS-CoV-2, are transmitted through aerosol particles produced by coughing, talking, and breathing. Medical procedures including tracheal intubation, extubation, dental work, and any procedure involving close contact with a patient's airways can increase exposure to infectious aerosol particles. This presents a significant risk for viral exposure of nearby healthcare workers during and following patient care. Previous studies have examined the effectiveness of plastic enclosures for trapping aerosol particles and protecting health-care workers. However, many of these enclosures are expensive or are burdensome for healthcare workers to work with. In this study, a low-cost plastic enclosure was designed to reduce aerosol spread and viral transmission during medical procedures, while also alleviating issues found in the design and use of other medical enclosures to contain aerosols. This enclosure is fabricated from clear polycarbonate for maximum visibility. A large single-side cutout provides health care providers with ease of access to the patient with a separate cutout for equipment access. A survey of medical providers in a local hospital network demonstrated their approval of the enclosure's ease of use and design. The enclosure with appropriate plastic covers reduced total escaped particle number concentrations (diameter > 0.01 μm) by over 93% at 8 cm away from all openings. Concentration decay experiments indicated that the enclosure without active suction should be left on the patient for 15–20 minutes following a tracheal manipulation to allow sufficient time for >90% of aerosol particles to settle upon interior surfaces. This decreases to 5 minutes when 30 LPM suction is applied. This enclosure is an inexpensive, easily implemented additional layer of protection that can be used to help contain infectious or otherwise potentially hazardous aerosol particles while providing access into the enclosure.

**Data Availability Statement:** The data underlying this study are available through Kilthub: https://doi.org/10.1184/R1/20514042.v1.

**Funding:** The authors received no specific funding for this work.

**Competing interests:** The authors have declared that no competing interests exist.

# Introduction

Infectious viral aerosol particles pose a serious threat to communities and individuals, especially health care providers, as exemplified by the ongoing SARS-CoV-2 pandemic. SARS-CoV-2 quickly spread globally and has proven persistent with more than 275 million cases and 5.3 million deaths (as of Dec. 20th, 2021) [1]. Rapid transmission primarily via aerosol particles (including the larger "droplets") led to overwhelmed hospitals and high exposure levels for healthcare providers. Studies have shown that low-income communities are disproportionately affected by the spread of SARS-CoV-2 [2]. This has led to the demand for increased protection for healthcare providers as well as their patients. There is now growing evidence that aerosols are a major transmission vector for other infectious diseases such as the common cold, seasonal influenza, and measles [3–6].

Viral transmission commonly occurs through three possible pathways: fomites on contaminated surfaces, contact with large airborne droplets ($> 5$ μm in particle diameter), and inhalation of smaller particles ($< 5$ μm) [7–10]. For SARS-CoV-2, the risk posed by fomites is the minor pathway for infection [11–13]. Aerosol particles are produced by coughing and sneezing, talking, and breathing [7, 9, 10, 14–16]. Particle size is the main controlling factor governing the transport, suspended lifetime, and thus exposure to aerosol particles.

The transmission vector of viruses through smaller aerosol particles is of grave concern due to their long airborne lifetime that allows these particles to travel long distances and disperse through HVAC systems [7, 17–19]. Particles containing the SARS-CoV-2 virus have been measured in hospitals [20–22]. In a clinical setting, transmission route models suggest that inhalation of aerosol particles, whether near an infectious patient or from dispersed aerosol particles, is a significant pathway for healthcare provider exposure and infection [23]. This aerosol exposure risk is increased at higher viral doses, which can be more pronounced during medical procedures where the patient cannot be masked, such as tracheal intubation, extubation, and suctioning [24, 25]. Studies show that not only are people more likely to get infected at higher viral doses, but they are also at increased risk of serious infection and disease [26–30].

Plastic barriers or hoods have been developed to place over the patient to contain produced aerosols [24, 25, 31–35]. A limitation is that the enclosure must have sizable openings to allow healthcare providers access to the patient. These openings also allow the aerosol particles to escape the enclosure, possibly also exposing the provider to a higher aerosol concentration in a localized area near the enclosure as the aerosol escapes through the smaller opening [24]. Previous enclosure designs have included two small hand holes for patient access, but this limits a health care provider's hand movement within the enclosure [36, 37]. The effectiveness of these barriers and feasibility of implementation in a medical setting is variable, with more effective measures such as active ventilation and/or filtration designed to reduce the risk of high particle concentrations near the provider but requiring greater infrastructure to operate [24]. The need for an additional layer of PPE provided by an enclosure has been questioned along with the restricted range of motion it incurs for the providers, and a patient's level of comfort within such an enclosure [38].

Evidence suggests that current levels of personal protective equipment worn by providers are adequate to meet the needs of medical staff in preventing the spread of infectious aerosol particles, particularly during extubation, intubation, and suctioning when new aerosol generation is not above background human emission levels [38–41]. However, this assumes that providers have ready access to all the necessary PPE, that it is properly functioning, and can be changed on a regular basis. It is not always possible to have sufficient PPE in all medical settings especially in times of crisis. Recent events have shown that even high-income nations with high-quality medical facilities can struggle to find the appropriate PPE at the onset of a

pandemic with medical staff resorting to garbage bags as gowns and reusing one-time use N95 respirators and face shields. The reusability of this enclosure design that we evaluate here, its ability to be sealed with easily sourced plastic films, and its effectiveness in trapping particles even without suction indicate utility even in resource-limited communities that are often hardest hit and in most need of critical medical resources. While the medical procedures targeted in the design of the enclosure are not generally considered aerosol-generating, that does not preclude the risk of transmission during their operation, nor does the design intent limit the enclosure's potential applicability to any procedure that requires an unmasked patient [39, 40].

Herein we present the results of aerosol experiments testing the performance of a simple enclosure barrier designed for preventing particle transmission during a simulated tracheal manipulation procedure, though its uses are not strictly limited to these sets of medical procedures. This barrier was designed to be inexpensive and easy to manufacture. Easily obtained malleable plastic sheet material was used to cover the openings to prevent escape of aerosols but allow provider good range of access to the patient. Additionally, observations presented here show no evidence that these plastic openings funnel escaping aerosol particles as was shown in previous literature [24]. Consideration of how best to implement this barrier with plastic coverings applied to minimize potential aerosol transmission is experimentally tested and discussed.

This enclosure design with coverings and the use of suction conforms to the revised FDA emergency use authorization released on August 21st, 2020 for such medical devices [42]. The current FDA guideline prohibits the use of passive enclosures. Yet, for enclosures with active suctioning (negative pressure), the FDA "believes that the known and potential benefits for emergency use of these devices, when used as authorized, continue to outweigh the known and potential risks and do not present public health or safety concerns at this time" [42]. This emergency use authorization is based on the understanding of the performance of the enclosures designs in wide use at the time of issuance. Studies such as this help provide a better understanding regarding how barrier design choices affect the performance of passive and active enclosure design for future consideration by regulatory bodies and healthcare providers. While the use of active filtration was necessary for other similar enclosures without added coverings to reduce aerosol escaping the enclosure, we saw no evidence that active filtration decreased particle leakage from this enclosure with coverings and the corresponding funneling of particles towards the providers [24]. Therefore, in situations where active filtration is unavailable, such as in rural and resource-limited communities, and remote or mobile operations, this enclosure will still provide an effective further layer to reduce the exposure to potentially hazardous aerosol. Active suctioning was found to greatly reduce the amount of time needed for aerosol loss via particle deposition to surfaces prior to removal of the enclosure. This simple inexpensive enclosure with plastic barrier coverings can help to reduce the spread of viral aerosol particles and other hazardous aerosol.

## Materials and methods

### Enclosure design

The enclosure is manufactured from clear polycarbonate by Magee Plastics Company (Warrendale, PA, USA) (Fig 1). The sides and top of the material are folded with the seams residing outside to promote easy disinfection of interior surfaces. The dimensions of the enclosure are 41.9 x 47.0 x 62.2 cm for a total volume of 122.5 L. As seen in Fig 1, three openings are cut into the sides of the enclosure to allow access to the patient during medical procedures. One large opening to allow easy access to the patient by the provider, one on the opposite side to accommodate the body of the patient, and one small opening on the side for the entry of medical

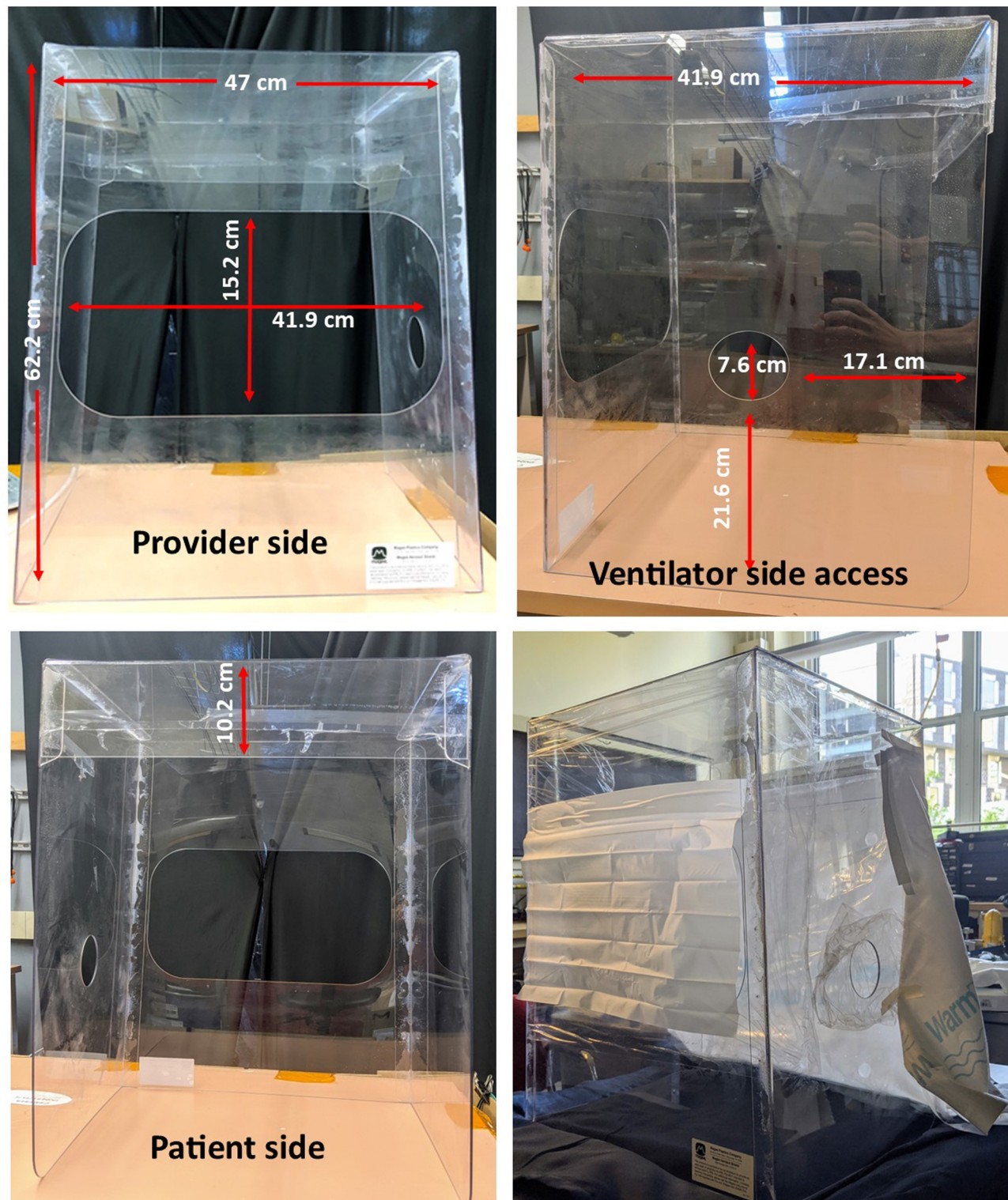

**Fig 1. Design and dimensions of the enclosure: A schematic of the enclosure showing the three cutout openings.** One on the cephalad (provider) side to provide access for the healthcare worker's hands, a second on the caudal (patient) side to allow room for the patient's torso, and a third on the lateral side to enable ventilator and other tubing access. The provider access cutout is large to maximize mobility.

equipment and tubing. The provider opening extends most of the width of the enclosure greatly reducing restrictions on the provider's arm movements compared to the common two-hole design.

Various disposable plastic sealing methods for the side and healthcare provider (front) openings were evaluated for their effect on the aerosol containment performance of the enclosure. Sealing methods were chosen based on their ease of procurement and use. Materials tested included Cling Wrap (Glad), Press'n Seal (Glad), Steri 1000 Drape (3M), and 50.8 cm wide furniture stretch wrap (Goodwrap). Further details on the make-up of the plastic covers are included in the supplemental information. Two cross-pattern (+) hand holes, 8–10 cm in diameter, were cut into the plastic sheet covering placed over the provider side opening. In some experiments, a Steri-Drape was placed over a layer of either Cling Wrap, Press 'n Seal, or furniture wrap. In these experiments, the Steri-Drape was not sealed on the bottom and hand holes were not cut into this top layer that was pushed up by the hands to access the patient. Aerosol instruments measured concentrations outside the enclosure with the sampling lines entering the side hole for internal measurements; the side hole was covered with one layer of the plastic being tested. For all experiments, the large patient-side opening was covered using material from a WarmTouch™ upper body blanket (Covidien) that was composed of an impermeable plastic barrier and a layer of cloth to prevent the movement of the thin plastic covering underneath. Further details can be found in S1 Appendix.

## Particle generation

Particles were generated from a 1 mg/mL ammonium sulfate aqueous solution. Several nebulizers were used to generate aerosol particles that can reach up to the diameter and velocity of particles produced by breathing or coughing (size distributions shown in S1 Fig). A Micro Air medical nebulizer (Omron) was used to produce smaller aerosol particles with lower ejection velocity. The medical nebulizer produced two modes in the particle number size distribution centered at 0.18 and 0.6 μm diameter. The large mode is similar to the lower aerosol size range exhaled during normal breathing [43]. In addition, this nebulizer is a self-contained propellant system similar to human breathing, eliminating positive pressure interference within the enclosure. To produce a size distribution and particle velocity more representative of a cough (i.e., particles with velocities upwards of 10 m/s), a Paasche Talon Airbrush was used and produced an aerosol number size distribution with modes at 0.2, 0.5, and a long tail stretching out past 3 μm [43]. The Airbrush was chosen based upon studies demonstrating that it generates particles similar in size to evaporated droplet nuclei generated by human coughing (0.74–2.12 μm) [43–46]. In all cases, the Paasche airbrush was pointed upwards at an angle of 60–75 degrees above horizontal, while the Omron medical nebulizer ejected particles vertically. Both nebulizers emitted particles 12–17 cm above surface level.

The longest period of particle generation, three minutes, was determined to increase relative humidity in the enclosure by 5 ± 2%. It is unlikely that this increase in relative humidity would have a significant impact on particle behavior over the minutes-timescale relevant to lifetime of particles in the enclosure when active suction is applied. Overall relative humidity varied by day but was within a normal "comfortable" range (~30–70%) for climate controlled indoor environments.

Generated particle concentrations were significantly higher than those observed by human coughing (S2 Fig). This was done to simulate 'worst-case' scenarios and amplify the aerosol signal that escaped the enclosure, facilitating reliable measurements since realistic coughing and tracheal operations release aerosol concentrations that are difficult to measure [40, 47]. The over-abundance of smaller particles in the hundreds of nanometers diameter range was a

result of the nebulizers used, but it was not seen as a limitation for this study. These particles have less inertia than larger particles and are better able to avoid impacting on obstacles and thus provide a more challenging test scenario of the enclosure's ability to prevent particle escape. Generating aerosol that more accurately represents the total emissions and particle size produced during human coughing, or during intubation and extubation procedures, would have created conditions in the enclosure that were less likely to lead to particle escape. Measurements were taken around the entire enclosure to find the most likely places for particle escape. Placing the copper sampling line inlet in areas most prone to particle escape also increased measured aerosol signal (S3 Fig). The sampling line was placed near the edges of the plastic coverings to determine the maximum particle leakage from the box.

## Sampling conditions

In some experiments, suction was applied at 15 or 30 LPM via a hose inserted through the side hole and sufficiently away from the nebulizer to not affect particle dispersal. Aerosol particle instruments applied a suction of 0.3 LPM or 3.3 LPM depending on the variables being tested. More information can be found in the supplemental information.

In some experiments, an investigator's hands wearing nitrile gloves were inserted through the hand holes cut in the front covering, moved within the enclosure for one minute, and then withdrawn to mimic motions and stresses placed on the enclosure during field use. In a few experiments, an aqueous solution of fluorescent fluorescein salt (Sigma-Aldrich) was nebulized to visually examine for areas of aerosol escape and deposition patterns within the enclosure. Deposited particles were illuminated via fluorescence using a blacklight.

## Instrumentation

Two condensation particle counters were the primary instruments used in this study (CPCs, TSI, 3772 and 3775). The CPCs have a 50% cut off at 10 nm (3772) and 4 nm (3775). They measured in one second or ten second integration intervals depending on the test and all data reported in this study was collected in particle number per $cm^3$ and then normalized by dividing maximum external concentration by maximum internal concentration. Further instrumentation was used to confirm results and gather size distributions, as described in the supplemental information. Tubing connected to the CPC sampling ports was placed in areas deemed most likely to leak during data collection. These areas included near the hand holes, near the side hole, or along the base at the back.

## Statistical methods

All experimental data reported were the result of four or five replicates. Statistical analysis was conducted in Matlab 2019a with built-in functions as discussed below. The Games-Howell test was a separate script to which one edit was made to correct an error in the original code [48].

## Results

The enclosure was subjected to use by medical staff in Allegheny Health Network hospitals. The medical staff who performed tracheal intubation and extubation procedures were surveyed on the patient access, visibility, and ergonomics of the enclosure during procedure (n = 39). The results are summarized in Table 1 with more detailed results and survey questions presented in the supplemental information. In all categories surveyed 90% or more of medical staff found the access, visibility, and the ergonomics of the enclosure agreeable or better. This contrasts with other enclosures where participants found their movement limited [38].

**Table 1. Survey of medical staff performing intubation/extubation with enclosure on mannequins (n = 39).**

|  | Strongly agree | Agree | Neutral | Disagree | Strongly Disagree |
|---|---|---|---|---|---|
| Safe endotracheal suctioning procedure | 56% | 36% | 8% | 0% | 0% |
| Easy equipment access | 46% | 54% | 0% | 0% | 0% |
| Good visibility | 74% | 23% | 3% | 0% | 0% |
| Acceptable Ergonomics | 51% | 43% | 3% | 3% | 0% |
| Effective helper access | 64% | 33% | 3% | 0% | 0% |

Laboratory experiments were conducted to determine the optimal covering configuration. To generate aerosol in the enclosure, a constant stream of particles was sprayed for 30 seconds with the Paasch airbrush. Several plastic covering configurations including a control set with no material covering the openings, one layer of plastic coverings, and two layers of plastic coverings on the front opening were examined. One layer of plastic was maintained on the side-hole even when dual covers were used on the front. Comparisons of aerosol concentrations inside versus outside the enclosure were made by taking the ratio of the maximum outside to maximum inside concentration to account for non-uniform mixing in the enclosure and to capture the most extreme cases of particle leakage from the enclosure. The maximum values being compared do not necessarily represent the same time in the run as there is a lag between when the particles are inside the enclosure, and when they have escaped the enclosure. Additionally, measurements were taken for long periods of time after generating aerosol to confirm that particles were contained and not slowly escaping over time.

To test the enclosure's ability to reduce exposure to a high viral dose, experiments compared the maximum concentrations of particles both inside and outside of the enclosure. The observed number concentrations of 0.01–10 μm particles inside and 8 cm away from the enclosure without plastic covers on the provider side indicated that the enclosure trapped up to 70 ± 11% of generated particles with 30 ± 10% escaping into the room (Fig 2). A single layer covering of furniture wrap increased the aerosol trapping efficiency to 86 ± 6%. The addition of a Steri-Drape covering over the furniture wrap proved effective: > 97 ± 3% of the aerosol particles were contained inside the enclosure. Steri-Drape placed over a layer of Press'n Seal was also tested with a trapping efficiency of 99 ± 1%. Adding 15 LPM suction to the Steri-Drape plus furniture wrap configuration reduced external aerosol concentrations by 98 ± 3% over internal concentrations. Summary of all statistical analysis can be found in the supplemental information.

These results indicate that plastic sheet covers over the openings of the enclosure lead to significant reductions in peak aerosol concentrations that escape the enclosure (one-way ANOVA p = <0.001). A Games-Howell test indicated a significant difference between no cover, 1-layer cover, and 2-layer covers. However, no significant difference between the different types of 2-layer configurations, with or without applied suction, was determined. Thus, the addition of plastic coverings reduces the peak concentrations of aerosol outside the enclosure.

Measurements were conducted at 8 cm, 15 cm, and 30 cm away from the openings to determine at what distance healthcare providers and equipment need to be positioned from the enclosure to minimize viral dose exposure from escaping particles. The medical nebulizer was used in the enclosure to generate particles for two minutes with no suction applied. Fig 3 illustrates the fraction of particles larger than 0.01 μm that escaped from the enclosure with either no coverings (Fig 3A) or Cling Wrap plus Steri-Drape (Fig 3B) on the openings. Two-sample t-tests were conducted at each measurement distance between covered and uncovered configurations followed by a Bonferroni correction to determine significance for multiple comparisons.

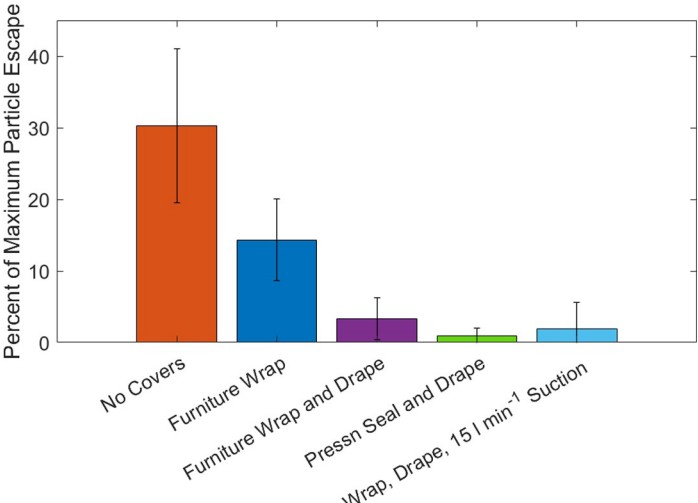

**Fig 2. Effectiveness of different enclosure covering approaches and materials: Ratio of the maximum aerosol number concentration measured inside vs. measured outside the enclosure for particles larger than 0.1 micrometers when various types of one-layer of plastic covering with or without an added Steri-Drape layer were placed over the front and side holes in the enclosure.**

There was a considerable reduction in total particle concentration at 8 cm (67 ± 12% no covers versus 7.6 ± 9% with covers, p = <0.001) from the enclosure. After the Bonferroni correction, the differences at 15 cm and 30 cm were not statistically significant. This discrepancy is likely a result of the large variability in measured particle concentrations at distances far from the uncovered enclosure. This is evident by the more than doubling in the standard deviation from measurements conducted at 8 cm compared to 15 cm. There was not enough particle leakage measured without a cover on the side hole to determine if there was a significant difference between cover and no cover at any distance for the side hole.

A simulated cough experiment using the airbrush determined the time required for particles inside the enclosure to settle onto interior surfaces. Air was pulsed through the airbrush in three 1-second bursts to simulate a cough. Fig 4A shows the time for particle number concentrations inside the enclosure to decrease by 90% from the peak concentration following the

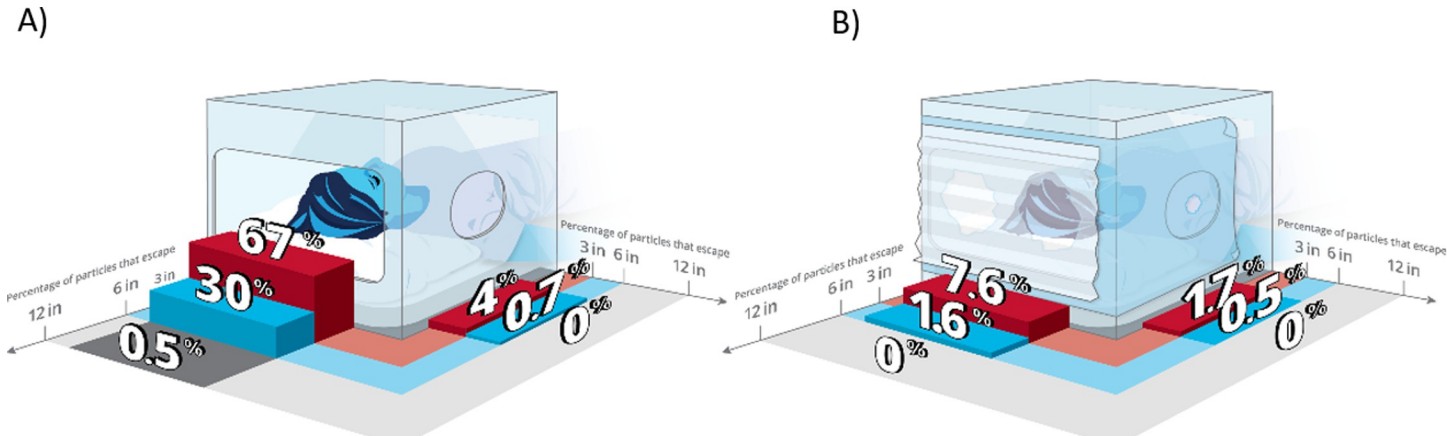

**Fig 3.** Aerosol particle concentrations as a function of distance from enclosure: Outside particle concentration compared to inside particle number concentration as a function of distance for enclosure (A) without plastic coverings vs. (B) enclosure with dual coverings of furniture wrap and Steri-Drape.

simulated cough event. With no coverings, this was achieved in roughly four minutes, likely driven by the rapid escape of particles from the enclosure. Adding two layers of covers increased the estimated 90% reduction time inside the enclosure to 14 ± 6 min. Particle loss of 80% was seen at 5 ± 3 min and 95% loss at 22 ± 12 min. Once suction was applied for the dual-covered enclosure, as shown in Fig 4B, the rate of particle loss was observed to greatly increase as suction flow rates were raised. The particle lifetime decreased from 7.1 ± 1 min without suction to 2.0 ± 0.2 min with the application of 30 LPM suction. This further indicates that the enclosure with two layers of coverings is effectively containing the aerosol such that the aerosol loss is now driven by deposition and settling to interior surfaces when suction is not applied.

To determine whether particles were depositing inside or around the enclosure's openings, and where, experiments with fluorescent particles were conducted. Results indicated no particle leakage in the visible particle size range (approx. 50 μm) from the enclosure without suction (S5 Fig). Most particle deposition was located on the bottom of the enclosure or directly in front of the spray nozzle. A fine film, noticeable when touched, was formed on the sides where suspended particles deposited onto the walls. This film was not observable prior to being agglomerated together through wiping, indicating that some particles smaller than the visible size range are also being trapped and deposited, likely through wall deposition and electrostatic attraction to the plastic walls of the enclosure.

Typical airway procedures require the provider's hands to reach into and out of the front opening of the enclosure, potentially jostling the device and leading to the increased escape of aerosol particles. Experiments were conducted to determine whether hand movements altered the number of aerosol particles that escaped the enclosure when covered with furniture wrap and a Steri-Drape. Hands were placed inside the holes cut into the furniture wrap cover either shortly before or shortly after a simulated cough event. Hands were moved while inside the enclosure to simulate a healthcare provider performing a procedure. After 1 minute the hands were removed from the enclosure. The results indicate no statistically significant increase in outside particle concentrations for either approach (S4 Fig).

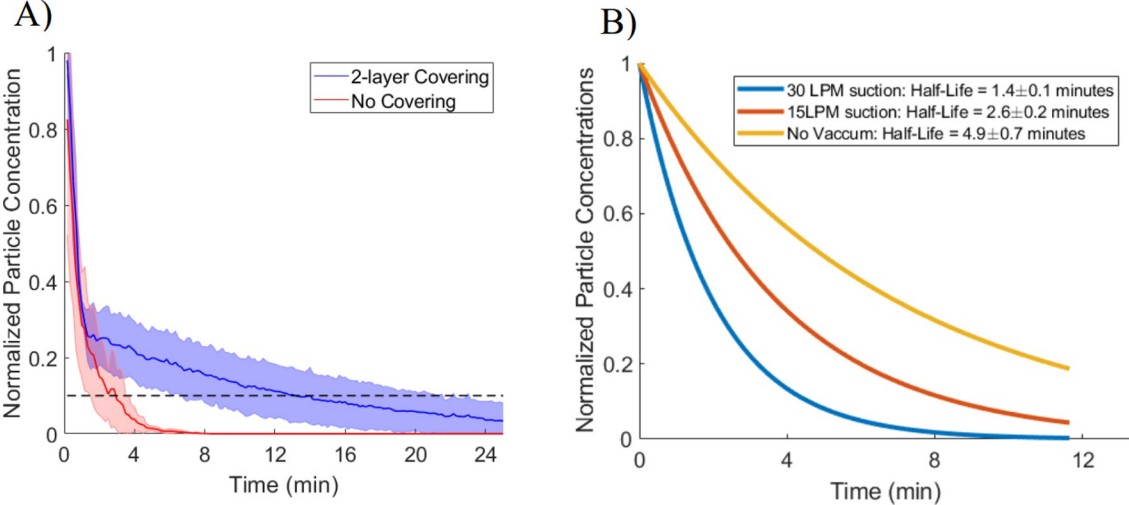

**Fig 4.** Particle concentration decay in the enclosure as a function of time and suction: (A) Average decay rate curves for aerosol particle number concentrations within the enclosure following a particle generation event. The no cover condition kept the provider and ventilator access holes open. The 2-layer cover condition used the furniture wrap and Steri-Drape. Shaded region shows the standard deviation from 3 replicates. The black dashed line indicates 90% particle loss. (B) Comparison of average decay rates for aerosol particle concentration when suction is applied to the enclosure. All three conditions involve a dual-covered enclosure with furniture wrap and Steri-Drape. The patient side was covered for all experiments.

Other field testing, such as agitating the enclosure by mildly shaking it, with and without hands inside the box, was conducted to simulate real-world conditions of using this enclosure. The measurements suggested no significant increase in outside particle concentrations (S4 Fig). This indicates that furniture wrap adheres well around the wrists and forearms of the provider. It also indicates that the dual-cover system can withstand general use while maintaining aerosol trapping performance within a medical setting with the associated mild bumps and movements during typical use.

## Discussion

The single large cutout for the provider's hands imparts a significant advantage over enclosure designs that have separate holes for each hand. The plastic covering with cut-in hand holes can bend and flex with the movement of the arms, minimizing the restrictiveness often experienced by users of other enclosure designs [36, 38]. This will provide medical professionals with much easier access to patients while using this enclosure. Additionally, there was no significant increase in aerosol particles escaping the box when tested under simulated real-world use conditions.

The measurements show that one-layer of plastic covering was not the most effective approach for trapping particles inside the enclosure. Adding a Steri-Drape on top of the first layer of plastic to create a two-layer covering was observed to be effective with more than $97 \pm 3\%$ of total aerosol particles larger than 0.01 μm contained within the enclosure. However, Glad Cling Wrap in the two-layer seal configuration proved to be less effective (not shown) compared to furniture wrap. There are two potential reasons for this result. First, the Cling Wrap covering was observed to electrostatically repel the Steri-Drape layer, creating a larger gap between the hand holes in the first layer and the drape that lies over these holes. Second, this material was more fragile than the furniture wrap and tore around the hand holes during use. The Press'n Seal covering appeared to electrostatically attract the Steri-Drape layer, potentially reducing opportunities for particles to escape through the cut hand holes. Both plastics showed reduced tearing at the hand holes during use. The furniture wrap combined with the Steri-Drape proved to be easier to use than the Press'n Seal combined with the Steri-Drape because of its better transparency and self-adhesive properties which enabled it to be secured to the enclosure without extra adhesive. Therefore, we recommend furniture wrap combined with the Steri-Drape to cover openings on the provider side of the enclosure. A single layer of wrap appears to be sufficient for the side hole. In a more general setting where these brands are not available, any adequately thick low-density polyethylene plastic sheeting should suffice, with high opacity and self-adhesive properties desirable.

The size of the enclosure and limited aerosol measurement instrumentation made monitoring all possible leak points impossible. However, care was taken to monitor places most likely to yield particle leakage during normal use (S3 Fig). These areas were where two materials met but could not be sealed with tape due to the necessity of access. At the primary sampling range of 8 cm, the air intake of the instruments would draw in many particles, particularly for the smaller particles which are more likely to escape. The strength of this draw diminishes rapidly with increasing distances and can help account for the decreasing measured concentrations at greater distances as particles had more volume to rapidly disperse and avoid detection.

The enclosure with two layers of covers applied over the front opening with no suction applied should remain over the patient for at least 15 minutes after the last particle generation event to achieve 90% reduction in particle concentrations within the enclosure. The addition of suction at 30 LPM and 15 LPM reduces this timescale to five and nine minutes, respectively. Particle losses within the enclosure are driven by several mechanisms depending on particle

diameter. Larger particles will inertially impact the enclosure if ejected with enough force or gravitationally settle to the bottom. Smaller particles are likely lost by an interception with internal surfaces as they circulate and diffuse within the enclosure, attracted via electrostatic forces, or removed by the suction flow. Particles remain larger for longer at higher relative humidity. This would increase the fraction of particles lost to gravitational settling, though this study did not investigate the extent of this impact [49]. Particles do not become re-aerosolized once they have deposited on a surface, and thus there is little concern of re-aerosolization when the enclosure is removed from the patient following the deposition period. Care should be taken when sterilizing the enclosure after use as fomites would be a concern for transmission. A sheet can be placed over the patient to collect most of the aerosol that settles at the bottom and then either disposed of or handled as contaminated and sterilized.

Proper sealing of the enclosure is critical to obtain high levels of particle retention within the enclosure. One advantage of the 50.8 cm plastic sealant is that it provides ample coverage around the front of the enclosure to minimize any chance of particle leakage through the edges. The users should take care to tuck in or seal with tape any other areas where particle leakage may occur particularly around the patient's body or through the side port. Extra-long and heavy material on the back opening allow this to be achieved more effectively. The experiments conducted with this enclosure captured the dynamics of particles of diameters similar to those produced by coughing and breathing. Both the airbrush and medical nebulizer generated particle sizes like those produced by passive respiration, speaking, and coughing, as well as even smaller particles ($< 0.3$ μm) [14, 43, 46]. The smaller particles are more mobile and difficult to trap than the larger particles produced by coughing. Therefore, these tests can be considered 'worst-case' scenarios containing higher concentrations of smaller particles that settle more slowly than would be present in a clinical situation. Nevertheless, we still observe drastic reductions in particle concentrations with the dual layer of plastic coverings without a corresponding increase outside the enclosure. Thus, it is reasonable to conclude that this enclosure will increase protection for healthcare providers from exposure to large viral doses from the smaller exhaled particles as well as larger cough droplets. This is not to say that all smaller particles will be trapped, despite the visible evidence that at least some of the smaller particles are deposited on the interior surfaces. The lack of corresponding increase in exterior particle concentrations suggest that the total exposure to potentially infectious aerosol particles providers are subjected to will be minimal. This reduces the chance of infection and extent of illness incurred [26–30]. However, due to the risk posed by lingering or escaped particles, current standards of PPE should be maintained whenever possible.

The design and performance of two other intubation enclosures were reported [31, 32]. These enclosures utilized a suction device and HEPA filters to help prevent the spread of aerosolized particles, with the openings either uncovered or covered with a rubber septum [31, 32]. Phu et al. reported a 99% particle reduction in the 0.5–5 μm range, which is slightly higher than the 93–97% reduction reported here without suction [31]. This is likely due to our particle measurements extending down to smaller sizes (to 0.01 μm) because these smaller particles can more easily escape the enclosure [24]. The passive aspect of the intubation enclosure presented here provides a significant layer of protection to reduce the spread of potentially infectious aerosol without the need for active suction to achieve similar particle reductions outside the enclosure. The main benefit of adding active suction to the enclosure is to reduce suspended particle lifetimes in the enclosure, and it is suggested that suction is used wherever feasible. The FDA has provided an emergency use authorization for barriers such as these only with negative pressure applied and rescinded its earlier approval of passive barriers without active suction.

This study investigates the use of an effective, inexpensive, simple to use, and easily sterilizable enclosure that utilizes accessible disposable plastic covers. Survey evidence by medical

professionals indicates that the enclosure can be used without hindering their work. The large side openings with plastic covers enable easy access into the enclosure. Various methods and materials to seal the openings in the enclosure were examined. A two-layer covering on the provider-facing opening consisting of furniture wrap and Steri-Drape on top was found to be the optimal method to reduce aerosol number concentrations escaping the enclosure. The side access hole should be covered with a layer of furniture wrap, and the patient side with a layer of plastic and a layer of heavier fabric on top. Depending on the amount of suction applied, a 5 to 20-minute waiting period following the last particle generation event is required to allow 90% of the particles to settle or be suctioned out before the enclosure is removed.

## Supporting information

**S1 Appendix. Further discussion on covers, particle generation, instruments, and other considerations.**
(DOCX)

**S1 Table. Summary of statistical result for cover comparison.**
(DOCX)

**S2 Table. Complete survey results of medical professionals that had used the enclosure for simulated intubation/extubation procedures on mannequins at two different hospitals.**
(DOCX)

**S1 Fig. Small particle size distribution from nebulizers.**
(DOCX)

**S2 Fig. Comparison of total particle concentration generated from real and simulated coughs.**
(DOCX)

**S3 Fig. Sampling intake locations for enclosure.**
(DOCX)

**S4 Fig. Comparison of fraction of particle escape during enclosure agitation.**
(DOCX)

**S5 Fig. Images of fluorescent particle settling and escape visualization experiment.**
(DOCX)

## Acknowledgments

Thanks go to Highmark Health for enabling work to be conducted on this study. Additional thanks to Dr. Peter Freeman of the Department of Statistics and Data Science at Carnegie Mellon University for statistical analysis input, and Tim Kelly for figure design support.

## Author Contributions

**Conceptualization:** Howard B. Gutstein, John P. Lawrence, Keith Lejeune, Ryan C. Sullivan, Coty N. Jen.

**Data curation:** Luke W. Monroe.

**Formal analysis:** Luke W. Monroe, Jack S. Johnson.

**Investigation:** Luke W. Monroe, Jack S. Johnson.

**Methodology:** Howard B. Gutstein, John P. Lawrence, Keith Lejeune, Ryan C. Sullivan, Coty N. Jen.

**Project administration:** Ryan C. Sullivan, Coty N. Jen.

**Resources:** Howard B. Gutstein, John P. Lawrence, Keith Lejeune.

**Software:** Luke W. Monroe.

**Supervision:** Ryan C. Sullivan, Coty N. Jen.

**Validation:** Howard B. Gutstein, John P. Lawrence, Keith Lejeune.

**Visualization:** Luke W. Monroe, Jack S. Johnson.

**Writing – original draft:** Luke W. Monroe, Jack S. Johnson.

**Writing – review & editing:** Luke W. Monroe, Ryan C. Sullivan, Coty N. Jen.

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
