## [Decision Letter · Decision Letter 0]

24 Feb 2022

PONE-D-22-01962Preventing spread of aerosolized infectious particles during medical procedures: a lab-based analysis of an inexpensive plastic enclosurePLOS ONE

Dear Dr. Sullivan,

Thank you for submitting your manuscript to PLOS ONE. After careful consideration, we feel that it has merit but does not fully meet PLOS ONE’s publication criteria as it currently stands. Therefore, we invite you to submit a revised version of the manuscript that addresses the points raised during the review process.

In particular, the reviewers generally agree that -despite the interesting question underlying the manuscript- some more accurate studies and controls should be carried out. This is particularly important in consideration of the relevance of aerosols and of their potential role in virus transmission in these days. It is thus mandatory for a scientific environment to vehicle messages beyond any possible misinterpretation.

If you can address the requests from the reviewers, I would be pleased to revise the manuscript.

We look forward to receiving your revised manuscript.

Kind regards,

Giovanni Signore

Academic Editor

PLOS ONE

Journal Requirements:

Reviewers' comments:

Reviewer's Responses to Questions

**Comments to the Author**

1. Is the manuscript technically sound, and do the data support the conclusions?

Reviewer #1: No

Reviewer #2: Yes

Reviewer #3: Yes

2. Has the statistical analysis been performed appropriately and rigorously? 

Reviewer #1: No

Reviewer #2: Yes

Reviewer #3: Yes

3. Have the authors made all data underlying the findings in their manuscript fully available?

Reviewer #1: No

Reviewer #2: Yes

Reviewer #3: Yes

4. Is the manuscript presented in an intelligible fashion and written in standard English?

Reviewer #1: Yes

Reviewer #2: Yes

Reviewer #3: Yes

5. Review Comments to the Author

Reviewer #1: In this manuscript the authors have created a ‘low-cost’ patient enclosure, generated aerosol inside the enclosure (with a solution of aqueous ammonium sulfate, azelaic acid and sucrose in deionized water) using a nebuliser to simulate breathing and a paint spraying airbrush to simulate cough. The authors used two condensing particle counters (CPC), a DustTrak aerosol monitor and an aerodynamic particle sizer (APS) to measure particles. Aerosol was measured inside and outside the enclosure to determine a ratio. One CPC was used within the enclosure, the Dustrak was generally used inside the enclosure and the APS generally used outside the enclosure. Only CPC values are reported to be shown in the paper.

A 30 second application of the airbrush was performed inside the enclosure. The authors found having uncovered apertures allowed 30% +/- 10% of the aerosol to escape. The application of a single layer of plastic over the apertures reduced the ratio of aerosol escaping and the addition of a second layer of plastic further reduced the ratio of aerosol that escaped. 2 minutes of the nebuliser within the enclosure also demonstrated occluding the hole reduced the amount of aerosol that escaped.

The authors tested the effect of suction at 15LPM and 30LPM on aerosol clearance and found suction decreased the time for aerosol concentrations to reduce to 90% of the peak. Fluorescein was nebulized to visually examine for areas of aerosol escape and deposition patterns within the enclosure.

The authors also conducted a survey of the enclosure to assess medical staff acceptability (n=39 respondents in 2 healthcare settings). It is unclear how this was performed as the manuscript references to medical staff “who performed tracheal intubation and extubation procedures” but the survey questions indicate this was simulated with mannequins.

I have a number of fundamental concerns with this paper:

1. Evidence quantifying the amount of aerosol generated from intubation and extubation in a real clinical environment has demonstrated these procedures does not generate aerosol. As such there is no evidence aerosol containment devices are required for these procedures [1,2].

2. The references cited by the authors pertaining to aerosol generation with intubation, extubation and suctioning do not support their statement. Simpson et al simulated aerosols generated by healthy volunteers coughing with a nebuliser placed in front of them [3]. Lindsley et al used coughing and breathing simulators to generate aerosol [4], neither of which reflect real clinical practice for intubation, extubation or suctioning.

As referenced by the authors, breathing, speaking and coughing does generate aerosol so the clinical applicability of this enclosure would require it be used at all times when a patient with a respiratory infection is conscious.

3. Recent studies and a meta-analysis have demonstrated aerosol containment devices increase time to intubation, increase first time failure rate, are more likely to breach PPE and have potential to cause patient harm [5-8]

4. This is supported by reference 39 from the manuscript which states:

“The FDA is revoking the umbrella emergency use authorization (EUA) for passive protective barrier enclosures (those without negative pressure.) We have carefully reviewed and considered preliminary evidence showing that there is a potential for adverse events or complications when using these devices while treating patients who are known or suspected to have COVID-19.”[9]

A further statement by the Journal Anaesthesia referring to intubation boxes/enclosures states:

“these devices were produced outside the normal regulatory framework, and thus were never clinically tested or validated for effectiveness and safety. They were subsequently heavily promoted in the media and on social media. Yet despite this heavy promotion, no international guideline on personal protective equipment (PPE) has ever endorsed their use.” [10]

As such this aerosol containment device is neither required or validated and is likely to increase the risk of harm to both patients and healthcare workers.

Also:

• There is no costing data as to the low-cost element, no cleaning data as to the ability to clean/sterilise/disinfect and no data on how easy it is to construct despite. There is no patient acceptability data.

• There has been no attempt to “simulate an airway procedure”. By inserting an operator’s hands into the box and removing them does not reflect real world practice of airway manipulation.

• The manuscript states all data are reported “in particle number per cm-3” (line 172) however there are no particle concentrations reported throughout the manuscript or supplementary material.

• There is no data on the background aerosol concentrations or where the study was performed.

• It is unclear how many experiments were performed. Throughout the manuscript the authors report "in some experiments" or "in a few experiments".

• The sampling locations for each experiment are unclear, sampling was performed at a number of locations but it is not clear when or for which experiment. The authors report “near the hand holes, near the side hole, or along the base at the back” and “numerous measurements were taken around the entire enclosure”.

• The use of visible fluorescein is a poor surrogate marker for aerosol deposition due to the differences in the size range of visible fluorescein compared to aerosol.

• Although acknowledged by the authors, Figure S3 demonstrates the simulated aerosol in this study is not representative of that generated by a physiological cough (being 1-2 orders of magnitude greater).

• There is a mix of imperial and metric units throughout the manuscript and figures

• Data analysis has been performed with a mix of parametric and non-parametric tests without justification.

1. Brown J, Gregson FKA, Shrimpton A, et al. A quantitative evaluation of aerosol generation during tracheal intubation and extubation. Anaesthesia 2021; 76: 174-81.

2. Shrimpton AJ, Brown JM, Gregson FKA, et al. Quantitative evaluation of aerosol generation during manual facemask ventilation. Anaesthesia 2022; 77: 22-7.

3. Simpson JP, Wong DN, Verco L, Carter R, Dzidowski M, Chan PY. Measurement of airborne particle exposure during simulated tracheal intubation using various proposed aerosol containment devices during the COVID‐19 pandemic. Anaesthesia 2020; 75: 1587-95.

4. Lindsley WG, King WP, Thewlis RE, et al. Dispersion and Exposure to a Cough-Generated Aerosol in a Simulated Medical Examination Room. Journal of Occupational and Environmental Hygiene 2012; 9: 681-90.

5. Lim ZJ, Ponnapa Reddy M, Karalapillai D, Shekar K, Subramaniam A. Impact of an aerosol box on time to tracheal intubation: systematic review and meta-analysis. British Journal of Anaesthesia 2021; 126: e122-e5.

6. Sorbello M, Rosenblatt W, Hofmeyr R, Greif R, Urdaneta F. Aerosol boxes and barrier enclosures for airway management in COVID-19 patients: a scoping review and narrative synthesis. Br J Anaesth 2020; 125: 880-94.

7. Begley JL, Lavery KE, Nickson CP, Brewster DJ. The aerosol box for intubation in coronavirus disease 2019 patients: an in‐situ simulation crossover study. Anaesthesia 2020; 75: 1014-21.

8. Noor Azhar M, Bustam A, Poh K, et al. COVID-19 aerosol box as protection from droplet and aerosol contaminations in healthcare workers performing airway intubation: a randomised cross-over simulation study. Emergency Medicine Journal 2021; 38: 111-7.

9. FDA. FDA In Brief: FDA revokes emergency use authorization for protective barrier enclosures without negative pressure due to potential risks., 2021. https://www.fda.gov/news-events/fda-brief/fda-brief-fda-revokes-emergency-use-authorization-protective-barrier-enclosures-without-negative

10. Association of Anaesthetists. https://anaesthetists.org/Home/News-opinion/News/Aerosol-boxes-and-COVID-19

Reviewer #2: This manuscript (PONE-D-22-01962) reports measurements of aerosol escape from a plastic enclosure designed to mitigate risk of exposure to infectious respiratory aerosols and droplets generated during clinical procedures like intubation and extubation. The plastic enclosure consists of several openings to accommodate the patient, the provider, and relevant medical instrumentation. In some cases, these enclosures were open, whereas in others they were covered using a range of plastic coverings. Aerosol was generated inside the enclosure using different medical nebulizers and was measured in the ~0.1-10 μm size range using a suite of aerosol measurement equipment that included condensation particle counters, scanning mobility particle sizers, and aerodynamic particle sizers. Aerosol escape, quantified by the ratio of maximum number concentration outside and within the enclosure, and the decay rate of aerosol generated within the enclosure were used to infer the effectiveness of the apparatus for mitigating exposure to potentially infectious aerosols.

This manuscript reports on a type of apparatus that has had significant use during the COVID-19 pandemic. The measurements made are largely appropriate, and this work is timely. The work is within the scope of PLoS One and will be publishable once the comments below are satisfactorily addressed. Given the subject matter of this manuscript, it is worth noting that this reviewer’s expertise is in aerosols and aerosol generating procedures; this reviewer has no clinical expertise.

Comments:

1. Lines 31-32 (“should be left on the patient for 15-20 min”): This seems like a very long time and potentially impractical if this apparatus were widely used. More broadly, how do these decay rates compare to those typical of the clinical environment in which these procedures usually occur? In their revised manuscript, the authors should comment more explicitly on the practicality of the apparatus given the results of their study.

2. Line 206 (and following discussion of Fig. 3): For these measurements, presumably the concentration inside the apparatus was relatively homogeneous due to the enclosure, but external to the apparatus there would be substantial dispersion, which would result in very rapid decreases in number concentration with distance from the opening. The authors should discuss this complication and how it was considered in the experimental protocol and data analysis. Is it possible that the number concentrations measured at 8 cm from the box are simply an artefact of the sampling position and may significantly underestimate aerosol leakage? The authors’ revised manuscript should enhance their discussion of this measurement, as it is a critical aspect of the manuscript and requires more careful discussion.

3. Lines 213-214 (“trapping efficiency”): Is “trapping efficiency” the most appropriate word? The incorporation of suction just means that aerosol preferentially will follow the gas flow to the suction device, reducing the amount that escapes through the other openings.

4. Lines 238-239: The authors should consider whether it might be appropriate to do a control experiment to see how number concentration decays with distance from the aerosol generation in the absence of an enclosure. Ultimately, the authors are reporting a concentration at some distance, but that concentration necessarily will naturally reduce simply due to dispersion. While aerosol might be homogenously dispersed within the apparatus, that will not be the case in the surrounding room. The authors need to address this challenge more directly in their revised manuscript.

5. Lines 248-249: Would it make more sense to fit an exponential decay and report the time constant (similar to estimating an air change rate)? Ultimately, this appears to be an appropriate parameter to describe the decay of the aerosol concentration within the apparatus.

6. Line 268 (“visible size range”): To what particle sizes does “visible size range” roughly correspond?

7. Lines 340-342: The statements here overreach, as the outcome will depend critically on, for instance, infective dose. If the infective dose is very low, then the enclosure may not actually reduce chance of infection. However, it will reduce total exposure to potentially infective aerosol.

8. Figures S1 and S2: Are the size distributions in these two figures self-consistent? On first glance, the SMPS and APS size distributions look to be very different in the limited size range where there is overlap. However, this could be due to the very different number concentrations. The authors should compare these size distributions in more detail in their revised manuscript.

9. More generally, and related to size distributions, how did relative humidity in the chamber vary with time? If relative humidity is increasing with increasing nebulization time, the size distribution (though not the number concentration) may change substantially. In their revised manuscript, the authors should discuss in more detail the size distributions and stability of these distributions over time.

10. This manuscript is motivated by a desire to reduce clinician exposure during aerosol generating procedures. However, there is minimal discussion about advances made in quantifying aerosol generation during procedures like intubation (for instance, the series of papers from Brown, Shrimpton, and Pickering, e.g. https://doi.org/10.1111/anae.15292, https://doi.org/10.1111/anae.15599). The authors in their revision should place their work in the broader context of our current understanding of aerosol generating procedures.

Reviewer #3: Introduction

The introduction is well written and adequately describes the state of the science.

Methods

Line 147. The size distribution of aerosol/droplets produced by a cough are commonly reported to be much larger than those used in the study (e.g. https://royalsocietypublishing.org/doi/10.1098/rsif.2013.0560). Why was this very small size region used?

Line 137 The trajectory of an aerosol size is highly dependent on size, which itself is a by-product of both the starting solute composition and relative humidity. Both of these parameters are not reported in the main body of the manuscript.

Line 163 Using fluorescein to visualize the areas wherein the aerosol escapes and is deposited is problematic for a number of reasons. Only the largest droplets will settle out of the air and deposit on the surface. Additionally, the level of fluorescence will be proportional to the size of the droplets themselves (e.g. larger droplets will fluoresce more). Collectively, this measurement will provide only a limited understanding of the spread of only the largest aerosol droplets. While this may useful for fomite transmission, it tells very little about the aerosol that may be inhaled.

Line 172 Instruments were used to gather size information, while only the CPC data was provided in the main text. Surely more useful information is within the size dependant data. The size dependent data should be in the main text.

Line 177 Why wasn’t any statistical relevance calculated to demonstrated significance?

Results

Figure 1 A detailed drawing of where the aerosols are being sampled should be provided, as well as the location of the nebulizer used to produce the aerosol

Figure 2 If no covers are present, the concentration immediately outside of the cover, should be the same as inside the chamber. This speaks to the point I’ve raised above. Unless the reader knows where the samplers are, it’s difficult to interpret this data. Moreover, “No Covers” is not an adequate control to measure the utility of the box. Rather, no box present at all should be considered.

Figure 3 What is the elevation that the samples are taken?

Discussion

Line 317 How do the authors know that 90% of sub micron droplets will deposit in 15 minutes. Are they not simply reporting the loss of aerosol detection? If that is the case, where is the aerosol going? My concern is that if this box is used to limit the exposure of aerosol, the authors should have a clear understanding what is happening to all of the nebulized sample.

In order for something to be defined as disinfected, around 99.999% of the infectious species needs to be inactivated. In this study, no where near that level of aerosol removal has been demonstrated. The authors need to make this limitation clear to the reader, and provide adequate context in the discussion.

6. PLOS authors have the option to publish the peer review history of their article (what does this mean?). If published, this will include your full peer review and any attached files.

Reviewer #1: No

Reviewer #2: No

Reviewer #3: No

---

## [Author Response · Author response to Decision Letter 0]

30 May 2022

Please see the attached Response file.

---

## [Decision Letter · Decision Letter 1]

12 Jul 2022

PONE-D-22-01962R1Preventing spread of aerosolized infectious particles during medical procedures: a lab-based analysis of an inexpensive plastic enclosurePLOS ONE

Dear Dr. Sullivan,

Thank you for submitting your manuscript to PLOS ONE. After careful consideration, we feel that it has merit but does not fully meet PLOS ONE’s publication criteria as it currently stands. Therefore, we invite you to submit a revised version of the manuscript that addresses the points raised during the review process.As you can see, two reviewers recommended your article for publication. However, one of the three reviewers which were originally contacted was concerned about the ethical and safety implications of the proposed device. On the basis of these criticisms, I performed a careful evaluation of the manuscript, and I would personally suggest that any direct reference or recommendation to use in a real hospital environment is removed from the revised version (especially, but not exclusively, in the conclusion). While I appreciated the technical part of the manuscript, I agree that we should be very careful in promoting novel devices tailored for medical use until relevant studies are completed. Please submit your revised manuscript by Aug 26 2022 11:59PM. If you will need more time than this to complete your revisions, please reply to this message or contact the journal office at plosone@plos.org. Please include the following items when submitting your revised manuscript:A rebuttal letter that responds to each point raised by the academic editor and reviewer(s). You should upload this letter as a separate file labeled 'Response to Reviewers'.A marked-up copy of your manuscript that highlights changes made to the original version. You should upload this as a separate file labeled 'Revised Manuscript with Track Changes'.An unmarked version of your revised paper without tracked changes. You should upload this as a separate file labeled 'Manuscript'.If applicable, we recommend that you deposit your laboratory protocols in protocols.io to enhance the reproducibility of your results. Protocols.io assigns your protocol its own identifier (DOI) so that it can be cited independently in the future. For instructions see: https://journals.plos.org/plosone/s/submission-guidelines#loc-laboratory-protocols. Additionally, PLOS ONE offers an option for publishing peer-reviewed Lab Protocol articles, which describe protocols hosted on protocols.io. Read more information on sharing protocols at https://plos.org/protocols?utm_medium=editorial-email&utm_source=authorletters&utm_campaign=protocols.

We look forward to receiving your revised manuscript.

Kind regards,

Giovanni Signore

Academic Editor

PLOS ONE

Journal Requirements:

Reviewers' comments:

Reviewer's Responses to Questions

**Comments to the Author**

1. If the authors have adequately addressed your comments raised in a previous round of review and you feel that this manuscript is now acceptable for publication, you may indicate that here to bypass the “Comments to the Author” section, enter your conflict of interest statement in the “Confidential to Editor” section, and submit your "Accept" recommendation.

Reviewer #2: All comments have been addressed

Reviewer #3: All comments have been addressed

2. Is the manuscript technically sound, and do the data support the conclusions?

Reviewer #2: Yes

Reviewer #3: Yes

3. Has the statistical analysis been performed appropriately and rigorously? 

Reviewer #2: Yes

Reviewer #3: N/A

4. Have the authors made all data underlying the findings in their manuscript fully available?

Reviewer #2: Yes

Reviewer #3: Yes

5. Is the manuscript presented in an intelligible fashion and written in standard English?

Reviewer #2: (No Response)

Reviewer #3: Yes

6. Review Comments to the Author

Reviewer #2: I have reviewed the response to reviewer comments and the revised manuscript and judge that the authors have satisfactorily and materially addressed all reviewer comments in their revised manuscript. I am happy for this manuscript to be published.

Reviewer #3: The authors have made great efforts to extensively and adequately address each of the points I raised, as well as those of the other reviewers.

7. PLOS authors have the option to publish the peer review history of their article (what does this mean?). If published, this will include your full peer review and any attached files.

Reviewer #2: No

Reviewer #3: No

---

## [Author Response · Author response to Decision Letter 1]

1 Aug 2022

Please see Response file submitted.

---

## [Editor Report · Decision Letter 2]

4 Aug 2022

Preventing spread of aerosolized infectious particles during medical procedures: a lab-based analysis of an inexpensive plastic enclosure

PONE-D-22-01962R2

Dear Dr. Sullivan,

We’re pleased to inform you that your manuscript has been judged scientifically suitable for publication and will be formally accepted for publication once it meets all outstanding technical requirements.

Kind regards,

Giovanni Signore

Academic Editor

PLOS ONE

---

## [Editor Report · Acceptance letter]

30 Aug 2022

PONE-D-22-01962R2 

Preventing spread of aerosolized infectious particles during medical procedures: a lab-based analysis of an inexpensive plastic enclosure 

Dear Dr. Sullivan:

I'm pleased to inform you that your manuscript has been deemed suitable for publication in PLOS ONE. Congratulations! Your manuscript is now with our production department. 

Kind regards, 

on behalf of

Dr. Giovanni Signore 

Academic Editor

PLOS ONE